# Physico-Chemical Characterization and Biological Activities of a Digestate and a More Stabilized Digestate-Derived Compost from Agro-Waste

**DOI:** 10.3390/plants10020386

**Published:** 2021-02-18

**Authors:** Antonella Vitti, Hazem S. Elshafie, Giuseppina Logozzo, Stefania Marzario, Antonio Scopa, Ippolito Camele, Maria Nuzzaci

**Affiliations:** 1Department of Pharmacy, University of Salerno, Via Giovanni Paolo II, 132, 84084 Fisciano, Italy; avitti@unisa.it; 2School of Agricultural, Forestry, Food and Environmental Sciences, University of Basilicata, Viale dell’Ateneo Lucano 10, 85100 Potenza, Italy; hazem.elshafie@unibas.it (H.S.E.); giuseppina.logozzo@unibas.it (G.L.); stefy.85s@libero.it (S.M.); ippolito.camele@unibas.it (I.C.); maria.nuzzaci@unibas.it (M.N.)

**Keywords:** organic materials, plant pathogens, antifungal activity, phytotoxicity, genotoxicity

## Abstract

The excessive use of agricultural soils and the reduction in their organic matter, following circular economy and environmental sustainability concepts, determined a strong attention in considering composting as a preferred method for municipalities and industries to recycle organic by-products. Microorganisms degrade organic matter for producing CO_2_, water and energy, originating stable humus named compost. The current study analyzed the chemical composition of a cow slurry on-farm digestate and a more stabilized digestate-derived compost (DdC), along with their phytotoxic, genotoxic and antifungal activities. The chemical analysis showed that digestate cannot be an ideal amendment due to some non-acceptable characteristics. Biological assays showed that the digestate had phytotoxicity on the tested plants, whereas DdC did not induce a phytotoxic effect in both plants at the lowest dilution; hence, the latter was considered in subsequent analyses. The digestate and DdC induced significant antifungal activity against some tested fungi. DdC did not show genotoxic effect on *Vicia faba* using a micronuclei test. Soil treated with DdC (5 and 10%) induced damping-off suppression caused by *Fusarium solani* in tomato plants. The eco-physiological data indicated that DdC at 5–10% could increase the growth of tomato plants. In conclusion, DdC is eligible as a soil amendment and to strengthen the natural soil suppressiveness against *F. solani*.

## 1. Introduction

The concept of circular economy has redefined, for the entire society, the terms of long period territorial growth and environmental benefits, so determined by the transition of fossil compounds to renewable energy sources. Effects on economic, natural, and social capital, based principally on the recycling of products and waste materials, as well as direct effects on regenerating natural systems have occurred [1].

In agriculture, the application of crucial concepts in circular economy and sustainability involves different aspects, such as the ability to cover the needs of future generations, the efficient use of natural resources, energy and soil consumption, the recycling of materials [2], the fertility of soils, nutrients and wastes, as well as the effects on water bodies and climate change [3,4].

The reduction in organic matter (OM) and the excessive use of soil stimulate the necessity to seek new methods to increase the soil OM content through the utilization of recycled organic waste [5].

The practice of adding agricultural organic waste to the soil is consistent with environmental constraints since it is able to improve physical, chemical and microbiological soil properties, which may also improve crop nutrition [4]. On the other hand, organic waste may also contain pollutants, especially if it is improperly managed or applied to the soil [6].

The possibility of obtaining energy from waste products has increased the interest in the anaerobic digestion (AD) process of animal manure and crop residues. The biological transformation of the organic matrix gives two by-products: biogas for energetic use and the anaerobic digestate (ADG), usable as a fertilizer and/or amendment in agriculture [7,8]. Circular economy challenges concern the achievement of renewable energy by agro-food waste, with the minimum environmental impact, and an increment of competitiveness in agriculture [9].

Several studies have assessed the qualitative/quantitative modification of waste, which comprises agro-food biomasses, during the anaerobic process that can provide useful information to better qualify and interpret the nature and the digestate actions as an organic amendment [10,11,12,13]. The optimum fertilizing properties of ADG might be lost when it is neither completely stabilized nor contains biodegradable matter. In order to overcome such a problem, composting as a post-treatment of ADG has been proposed [14,15]. Meanwhile, the composting process can transform the digestate into a mature stable, safe, humus and nutritive compost, which provides the relevant advantage to maintain and improve environmental quality and conserve resources. Furthermore, compost represents an important source of organic matter and nutrients for agriculture, playing a crucial role in sustaining soil biodiversity [16] and the production of horticultural species when used as a component in the preparation of pot substrates [17]. Moreover, the microflora of the compost have significant antagonistic effects against several soil-borne phytopathogenic microorganisms, and hence they can play a vital role in controlling them [18,19,20].

Physico-chemical analysis of compost is limited to offer the real information about its compositions, where the heavy metals and organic waste could produce polluting molecules and induce phytotoxic and/or genotoxic effects in crops [21].

Broad bean (*Vicia faba* L., family *Fabaceae*) plants are largely used for the toxicity evaluation of different matrices as a sensitive organism to assess nuclear damages, chromosomal aberrations, gene mutations and micronuclei inductions [22,23,24,25]. During cell division, damage/ill-repaired cells cause micronuclei inductions and daughter cells become abnormal in terms of size [26]. Micronuclei tests on the root tip cells of broad bean plants have been used to identify different forms of environmental stress [27] and to assess the genotoxic effects of the environmental pollutants and chemicals [28].

Currently, tomato (*Solanum Lycopersicum* L., family *Solanaceae*) is one of the most widely consumed vegetables worldwide. Tomato crop production suffers due to climate changes occurring in several areas, especially in the Mediterranean region. In fact, the general increase in air temperature and/or the relevant reduction in precipitation can prejudice plant defensive mechanisms and increase the risk of illness through growth and physiology alteration of the host plant and also by modifying host-pathogen interactions [29]. In particular, soil-borne fungi, such as *Fusarium solani* and *Rhizoctonia solani*, represent the causal agents of damping-off and/or root rot diseases in tomato plants. These pathogens affect the seed germination and young seedlings, and they become the limiting factor in the production of both open fields and under greenhouse cultivation systems [30].

The current study aimed to assess the potential use as an organic amendment of a cow slurry digestate produced on-farm and/or a stabilized digestate-derived compost (DdC) obtained after 50 days by an anaerobic composting process, in self-built semi-batch reactors, assuming that the post-treatment by composting could determine a final product which is more usable for agricultural purposes.

In particular, we investigated: (i) the chemical composition of the digestate and compost; (ii) potential phytotoxicity to the germination and root elongation in gardencress (*Lepidium sativum* L., family *Brassicaceae*) and wild radish (*Raphanus raphanistrum* L., family *Brassicaceae*); (iii) the antifungal activity of the broth extracted from the digestate and DdC against some common phytopathogenic fungi; (iv) the genotoxicity effect of DdC mixed with growing soil on broad bean (*Vicia faba* L.) root tips; (v) the effect of compost on post-emergence infection damping-off caused by *F. solani*, together with pot germination tests and the study of eco-physiological parameters in tomato (*S. lycopersicum* var. *cerasiforme*).

## 2. Results and Discussion

The concepts of circular economy and environmental sustainability are very important due to the huge use of resources responsible for environmental contamination by xenobiotic compounds, microorganisms and all the waste produced by agro-industrial processes, as well as household activities. Therefore, it is considered of great interest to verify the possibility of satisfying the energy needs of the agro-industrial productive areas through the exploitation of renewable sources available on site, and their transformation into organic matter, such as digestate and DdC, to be used in agricultural systems.

### 2.1. Analysis of Digestate and Digestate-Derived Compost

The changes in temperature in the reactor producing DdC indicated that it reached its highest value of 51 °C after 3 days. Then, it started to decline and stabilized at 26 °C until the 50th day. pH values were 8.34 and 7.85 at the beginning and the end of the process, respectively (personal communication; data not shown).

Physico-chemical properties of on-farm digestate and DdC are reported in Table 1. The latter product was considered as a mixed composted amendment according to the Italian Legislative Decree 75/2010 about “Reorganization and Revision of the Discipline on Fertilizers”. The pH values were alkaline for the digestate (8.10) and DdC (7.88); however, they were acceptable in agriculture according to the Decree values of pH, which ranged between 6.0 and 8.5. The composted materials were considered in the evolution phase, as demonstrated by the low values of the carbon to nitrogen (C/N) ratio (4.5), while the values considered ideal for a completely mature compost are 15–30 [31]. This parameter indicated the presence of a high amount of available nitrogen in the case of compost, and this product also had a good amount of organic carbon (40% d.m.), as requested by the Decree (minimum value of 20%) with respect to the digestate, with a definitely lower value of 6.8% d.m. The digestate showed a low value of humic carbon, a higher moisture and a lower germination index than DdC, hence it could be acceptable as a possible amendment until the full maturation process, such as the obtained DdC.

Generally, agricultural soils can be contaminated by toxic trace metals and pathogenic microorganisms when unsuitable manure, animal waste, and decaying plant material derived from agricultural processes are used as soil conditioners [32]. In fact, heavy metals and pathogenic microorganisms are of special concern for the environment and agro-ecosystems, and could cause serious problems for soil biotic compartments and human health. Regarding the heavy metals, their amounts in the digestate and DdC were lower than the maximum admitted contents by the Italian Legislative Decree 75/2010, and therefore not decisive for inducing eventual abiotic stress (Table 1). On the other side, digestate samples contained a higher number of *E. coli*, confirming their need to be subjected to further maturation because they did not respond to the main characteristics of compost quality requirements proposed in national (L.D. 75/2010) and international (EU fertilizer decree for CE marked fertilizers, 1009/2019) regulations [33].

### 2.2. Phytotoxicity Assay

The phytotoxic effect of the digestate and DdC was performed on *L. sativum* and *R. raphanistrum* at different concentrations, and the results are listed in Table 2 and Table 3.

Luo et al. [34] explained that a GI equal to or higher than 80% indicates the absence of phytotoxic effects. The digestate sample reduced the GS and RE of both studied plants as the tested concentrations increased. At the same time, when used at 2.5%, both GS and RE were not significantly different compared to the control, except for the GS of radish. These findings are in agreement with Lencioni et al. [35], who reported that digestate concentrations ranging between 2 and 3% are able to stimulate the germination and early life stages. The same authors also stated that it is strongly advisable to avoid direct contact with germinating seeds to minimize the eventual phytotoxic effects of digestate. In fact, all tested concentrations of the digestate sample in our study determined GI values which were always significantly lower than the control and the threshold level (80%), thus indicating their phytotoxic effect (Table 2).

Additionally, the GI values of DdC decreased with the increase in the concentrations. Interestingly, at the lowest tested concentration (2.5%), DdC showed a GI not significantly different to the control and much higher than the threshold level (80%), thus indicating the absence of phytotoxicity on both tested species.

Generally, the potential phytotoxic nature of organic waste such as the digested sample and also mature composted organic materials is mainly due to the combination of several factors, such as high salinity, heavy metals and fatty acids, as reported by Delgado et al. [36]. The phytotoxicity effect of digestate, found also at a low concentration, could be explained by its higher pH, and perhaps also its salinity and lower values of organic carbon, humic and fulvic acids than DdC [37], as shown in Table 1.

### 2.3. Antifungal Activity

Regarding the digestate, the results showed that the highest significant activity was observed with both tested concentrations (100 and 50%) against *M. laxa*, *M. fructicola*, *M. fructigena, F. solani* and *V. dahliae* (Table 4). In addition, moderate fungicidal activity was observed against *R. solani* and *S. sclerotiorum* in the case of the lower concentration (50%). Furthermore, the concentration at 50% showed low activity against *A. flavus* and *A. ochraceus*.

Regarding the DdC, the results reported in Table 5 show that the highest significant antifungal activity was observed at the two tested concentrations against *M. laxa*, *F. solani* and *V*. *dahliae*, whereas there was moderate activity against *M. fructicola*.

The application of different types of compost to farmland has been reported to suppress several soil-borne diseases, including *Fusarium wilt* [38], *Pythium*, *Phytophthora*, *Verticillium* and *Rhizoctonia* [39]. This is due to the predominantly biological effects of compost, involving the activation of soil microorganisms and plant roots, improving the physical and chemical properties of soil, as well as of the nutritional status of plants [17,39].

### 2.4. Genotoxic Effect of DdC on Broad Bean Plants

The genotoxicity of DdC when mixed with soil at different percentages was detected in terms of frequencies of micronuclei. Generally, genotoxic effects are divided into three grades based on micronuclei frequencies: 0–1.5, no genotoxic effect of the analyzed soil sample; 1.5–2, light genotoxic effects; and above 2, heavy genotoxicity [27].

A preliminary observation of *V. faba* seedlings indicated a decreasing growth rate when DdC concentration increased in the tested soil samples from 0 to 20% (data not shown).

However, when assessing micronuclei, no differences were observed in the 5%, 10% and 20% soil samples in comparison to the control (0%) (Figure 1).

The obtained results are in agreement with El Fels et al. [40], who analyzed solid matrices of two mixtures with Palm-date waste and did not find any genotoxicity effect. This could be related to interfering phenomena of pollutants and organic matter of the compost and the soil, which affect the root bioavailability of contaminants, particularly metals. The analyzed DdC has no potential genotoxic effects, and therefore it could represent an important source of organic matter and nutrients for sustainable agriculture.

### 2.5. Effect of Compost on the Suppressive Activity Against F. solani and Eco-Physiological Parameters in Tomato Plants

The evaluation of DdC for its potential control of damping-off (DO) induced by *F. solani* in post-emergence *S. lycopersicum* seedlings was performed. In general, the post-emergence damping-off symptoms are observed until the 4th–6th week post-sowing [41]. Therefore, the compost suppressive effect against *F. solani*, as well as the eco-physiological parameters in tomato seedlings, was evaluated 30 days post-sowing in soil containing only the phytopathogen (F; control untreated) or *F. solani* and compost at different concentrations.

The effect of DdC on DO disease incidence was significantly affected by the percentage of compost added to the soil, as summarized in Table 6. In fact, compared with the non-treated soil, where DO was the highest, the reduction in DO was gradually reduced with the decreasing DdC concentration.

Favorable conditions to the development of *Fusarium* sp. and DO disease incidence, accordingly, were indicated in a higher soil pH with increased N levels, other that the pathogen species [42]. In the present study, the soil amended with 20% DdC is richer in N and has a higher pH than the soil at lower percentages of DdC, thus explaining the lack of suppression action.

In summary, our finding highlighted that the DdC used in the current study could be able to induce a suppressive action against *F. solani* even in an in vivo test. Yangui et al. [43] reported that the addition of olive mill wastewater to soil is able to enhance the disease suppressive effect against *F. solani* in tomato plants. To the best of our knowledge, the present study is the first report on using DdC for this purpose.

Figure 2 shows the number of tomato seeds germinated in soil treated with DdC at different concentrations, alone or in the presence of *F. solani*.

The significantly highest and lowest values of germinated seeds were observed in the soil untreated and not inoculated (C, control) and inoculated only with *F. solani*, respectively. The presence of the compost in the soil did not induce a significantly positive effect on seed germination when used alone, if compared with the control soil, in particular at the highest percentage of 20%. Interestingly, the DdC determined a significant and ameliorative number of germinated seeds in the soil also treated with the pathogen, with respect to the only inoculated control soil (F).

Recently, it was demonstrated a biological control activity against *F. solani*, inducing the DO disease in tomato plants, by using two bacterial filtrates, which provided high protection and an increasing tomato seed germination percentage [44]. Our finding suggested that the DdC has a composition of microbial species able to exert this same protective action.

Additionally, the induced effect was gradually better as the compost concentration used decreased, as already observed for the DO bioassay. The significant reduction in the seed germination induced by the highest concentration of the compost used could probably be attributed to an inhibition of water uptake necessary for germination linked to the high salinity of the compost, as demonstrated by Nasri et al. [45] for lettuce plants.

In Figure 3, DdC effects on the height, leaf number, and shoot and root fresh and dry weights are shown. By adding the compost alone, in terms of plant height (Figure 3a) and number of leaves (Figure 3b), only the lowest percentage of treatment determined more growth, compared with the diseased plants (F) and the control ones (C). When the compost was also in the presence of *F. Solani*, it induced at 5% and 10% an improvement in plant height, but not in the leaf number, neither at low percentage with respect to the diseased plants (F).

Regarding weights, the used DdC at all concentrations determined higher values than the control in shoot fresh and dry weights, but not significantly relevant in the latter (Figure 3c,d). On the contrary, only the fresh root weight significantly improved in the diseased plants, and only when the compost at 10% was used (10% + F), although higher values than control F plants in root fresh and dry weights were always observed with the 5 and 10% compost treatments. These findings are in agreement with other studies showing the ability of the compost to increase the availability of nutrients for the plant, therefore promoting the absorption of nutrients and the growth of the roots with the following increase in the dry weight of both shoots and roots [46,47]. In summary, our results suggested that the compost at 5–10% could be able to increase the growth and development of tomato plants.

## 3. Materials and Methods

### 3.1. Substrate and Procedures

Digestate was provided from the zoo-technical farm, located in Atella (Basilicata Region, Italy) (40°51′ N; 15°38′ E), that produces biogas derived from the anaerobic digestion of livestock effluent and agricultural waste. Samples of digestate were collected in sterile containers, transported to the laboratories at the University of Basilicata in Potenza in ice chests and stored in a refrigerator (4 °C) until use.

Wheat straw used in this experiment, shredded into small pieces, was obtained from a farm that cultivated a durum wheat landrace near Genzano di Lucania (Basilicata Region, Italy) (40°51′ N; 16°02′ E).

Compost was obtained in thermally isolated hand-made semi-batch reactors. Each reactor was cylindrical in shape, with a volume of 0.05 m^3^, projected and built with high-quality plastic materials, and equipped with the qualitative and quantitative parameters control systems. Temperature profile and pH values during the composting development were measured (data not shown).

Briefly, the semi-batch sets handmade of 12 continually stirred digesters were filled with 90% of ADG and 10% wheat straw, allowing them to evolve the reactions for 50 days. The agitation of composting substrates for 5 min every 3 h was applied to improve homogeneity and to provide aeration and temperature control. Each reactor was filled up to 35% of the total volume and kept constant throughout the experiment. Finally, the DdC was separated and air-dried for 4 days.

### 3.2. Sample Preparation

#### 3.2.1. Digestate

One hundred milliliters of digestate sample was vortexed for 5 min with 100 mL of a sterile physiological solution (NaCl) 0.85% in a volumetric flask. The mixture was then incubated in a rotary-incubator (Certomat^®^ HK., B. Braun Biotech International, Melsungen, Hessen, Germany) for 2 h at 25 °C, filtered using a sterile net and a 12.5 cm diameter and 75 ± 5 g/m^2^ filter paper (Whatman Grade 41, Darmstadt, Germany), and centrifuged at 13,000 rpm at 20 °C for 15 min. The final obtained volume was concentrated using a rotary evaporator (Heidolf WB 2000, Schwabach, Germany) at 150 rpm and 50 °C for 30 min until completely dry. The final sample was stored at 4 °C for the following antifungal and phytotoxicity assays.

#### 3.2.2. Digestate-Derived Compost

Dry compost sample (50 g) was suspended in 200 mL of distilled water in a volumetric flask and incubated in a rotary-incubator (shaker) for 24 h at 25 °C. The mixture was filtered using Whatman Grade 41. The compost sample was stored at 4 °C for the following biological assays.

### 3.3. Physico-Chemical Characterization of Digestate and Compost

Compost samples from each reactor were thoroughly mixed to achieve homogeneity. The compost and digestate were preventively dried at 105 °C for 12 h. Chemical composition of both digestate and compost was carried out with the application of method UNI 10780:1998. The content of the elements [Al, As, B, Ba, Be, Cd, Co, Cr(VI), Cu, Fe, Hg, Mn, Mo, Ni, Pb, Sn, V, Sb, Se, Zn, and Ti] was determined after mineralizing their residues in a mixture (1:12 *w*/*v*) of concentrated nitric and Perchloric acids (3:2 *v*/*v*) by using Inductively Coupled Plasma Optical Emission Spectrometry (ICP-OES, Perkin Elmer Optima 7300 DV). For the analyses, deionized Milli-Q grade water (Millipore Corp., Bedford, MA, USA) and chemicals of analytical grade were obtained from Merck (Darmstadt, Germany).

### 3.4. Phytotoxicity Test

A bioassay based on seed germination and root elongation was carried out to evaluate the phytotoxic effect of the digestate and compost on *L. sativum* and *R. raphanistrum* following the method reported by Ceglie et al. [17] and Mancini et al. [48].

Briefly, seeds were firstly sterilized by soaking them in 3% hydrogen peroxide (H_2_O_2_) solution for 1 min and then rinsing twice with sterile water. Ten seeds were transferred in deionized water (control), digestate or DdC solutions at 10, 5, and 2.5 g/100 mL and shaken gently for 2 h.

Ten seeds of each tested plant were placed separately on Petri dishes containing a filter paper (Ø 90 mm, Whatman No. 1, Darmstadt, Germany), wetted with 5 mL of sterile distilled water, digestate and compost solutions at above the mentioned concentrations and sealed by Parafilm^®^. All plates were incubated at 25 °C in darkness for 3 days. The experiment was conducted in triplicate and the root length of each seedling was measured. The Germination Index (GI) was calculated using Equation (1):GI (%) = [(GSt × REt) / (GSc × REc)] × 100,(1)
where GI is the germination index; GSt represents the average number of germinated treated seeds; REt is the average radical elongation for treated seeds; GSc is the average number of germinated seeds for the control; REc is the average radical elongation for the control.

### 3.5. In Vitro Antifungal Activity

#### 3.5.1. Fungal Isolates

Tested phytopathogenic fungi were stored at 4 °C as pure cultures and maintained in the mycotheca of the School of Agricultural, Forestry, Food and Environmental Sciences (SAFE), University of Basilicata (Potenza, Italy). The fungal species were cultured on Potato Dextrose Agar (PDA) at 24 ± 2 °C. The tested fungi were Rhizoctonia solani J.G. Kühn., Sclerotinia sclerotiorum (Lib.) de Bary, Aspergillus niger van Tieghem., Aspergillus flavus Link ex Gray, Aspergillus ochraceus, Botrytis cinerea Pers., Monilinia laxa (Aderhold and Ruhland), Monilinia fructicola (Winter) Honey, Monilinia fructigena (Aderhold and Ruhland), Fusarium solani (Martius) Saccardo and Verticillium dahliae Kleb.

#### 3.5.2. Fungicidal Assay

The fungicidal inhibitory activity of digestate and DdC samples was determined in vitro using the contact-phase method reported by Elshafie et al. [49]. Twenty microliters from each digestate and DdC at 100 and 50%, compared to Azoxystrobin (0.80 µL/mL) used as a control, was carefully applied over discs, and 0.5 cm diameter of fungal disks was inoculated in the center of each Petri dish containing PDA. All plates were incubated at 24 ± 2 °C for 4 days in darkness. The diameter of eventual inhibition zones was measured in mm.

### 3.6. Genotoxicity Test of DdC in Broad Bean Plants

Micronuclei tests of broad bean root tips were performed to detect the genotoxic effect of DdC when mixed with soil in different percentages. The above described DdC was mixed with a commercial growing soil (CGS) according to the following conditions:-0%: no compost (control);-5%: 950 g of CGS + 50 g of compost;-10%: 900 g of CGS + 100 g of compost;-20%: 800 g of CGS + 200 g of compost.

Four pots (Ø 6 cm) for each compost concentration were filled and micronuclei assay was carried out according to Kanaya et al.’s [50] protocol with minor modifications.

Broad beans were soaked for 18 h in distilled water at room temperature. The beans were then transferred in the filled pots (two beans for each pot) and allowed to germinate for 4 days at 20 ± 1 °C in the dark. At this time, the seedlings had grown 3–5 cm long primary roots. The primary root tip (1 cm) was carefully cut off and the seedlings were subsequently re-transferred in their corresponding pot at 20 ± 1 °C in darkness to allow the development of lateral roots. After 4 days, the lateral roots were left to grow to a suitable length (1–2 cm) to be used in the test.

The roots were fixed in Carnoy’s solution (ethanol and glacial acetic acid, 3:1 *v*/*v*) and kept at 4 °C for 24 h. Root tips were washed with distilled water, hydrolyzed in 1N HCl at 60 °C for 8 min and then Feulgen stained for 4 h. Squash preparations were produced in 45% acetic acid and 1000 cells per slide were scored and observed under an optical Microscope (Axioskop, Zeiss, Germany) at 40× resolution. The eventual genotoxicity effect of the DdC was determined based on the micronuclei frequencies per 1000 cells (MN‰), and the Pollution Index (PI) in each slide was calculated using Equation (2):PI = MN‰ of the sample / MN‰ of the negative control,(2)

### 3.7. Effect of DdC on Ecophysiological Parameters and Suppressive Activity Against F. solani in Tomato Plants

#### 3.7.1. Materials and Fungal Isolates

Compost used for the analysis of suppressive activity against *F. solani* in a pot assay in tomato (*S. lycopersicum* var. cerasiforme) was prepared by mixing the DdC described above with a sterile CGS (control) at 5, 10 and 20%, according to the same conditions used for the genotoxicity assay.

*F. solani* was isolated from tomato on PDA at 25 °C. The fungal isolate was preliminarily tested for pathogenicity on tomato and showed its typical behavior. The isolate was conserved in the fungal collection of SAFE. Pathogen inoculum was applied by using a suspension of spores at the concentration 1.8 × 10^6^ UFC/g of soil substrate.

#### 3.7.2. Experimental Setup

Tomato seeds were sterilized using 1% Na-hypochlorite solution for 1 min and then rinsed with sterile water before imbibition on moist filter paper at 4 °C for 24 h in darkness. Ten seeds were directly sown in each pot filled with autoclaved (at 121 °C for 30 min on two consecutive days) soil substrates pre-inoculated one week before with *F. solani*. Throughout the experiment, plants were kept in a growth chamber with a 16 h photoperiod, at 26/23 °C (day/night), and the plants were watered individually with tap water as needed.

The experimental design included eight conditions and three replications: (1) untreated and healthy plants; (2) plants inoculated with *F. solani*; (3–5) plants treated with DdC at 5%, 10% and 20%; (6–8) plants treated with DdC at 5%, 10% and 20% and inoculated with the pathogen. Seed germination was monitored by counting the number of emerged seedlings for each pot. After 30 days from fungal inoculation, the effect of DdC on post-emergence damping-off caused by *F. solani* on emerged tomato seedlings was evaluated using Equation (3), as described by Veeken et al. [51]:%DO = [(HPo − HPi) / HPo] × 100,(3)
where HPo is the number of healthy plants in the untreated experimental condition and HPi is the number of healthy plants in the treated compost and *F. solani* inoculated condition.

Additionally, the height of the plants, the number of leaves per plant, and the fresh and dry (after being oven-dried at 60 °C for 48 h) weights of shoots and roots for five randomly chosen plants were determined.

### 3.8. Statistical Analysis

All experiments were carried out in triplicate. Data of each experiment were expressed as the mean ± SDs, and statistically analyzed using SPSS software (SPSS; version 13.0, Prentice Hall, Chicago, IL, USA, 2004), followed by comparison of means (one or two-way ANOVA) using Tukey’s multiple comparison test, at the significance level of *p* < 0.05 or *p* < 0.01.

## 4. Conclusions

In an optical of sustainability and circular economy, the current study assessed the potential use as an organic amendment of a cow slurry digestate produced on-farm and DdC obtained after 50 days by an anaerobic composting process in self-built semi-batch reactors.

The study indicated that the digestate could be not considered as a good amendment, or it needed a further maturation process, as then better obtained by the DdC.

In fact, the digestate post-treatment by composting resulted in a product suitable as an important source of organic matter and nutrients.

In addition, DdC determined in vitro antifungal activity against some common serious phytopathogenic fungi, in vivo suppression action against *F. solani* by reducing DO in tomato, as well as the ability to increase the growth and development of tomato plants when used at a low percentage (5–10%).

In conclusion, the current study proved that a further maturation process of on-farm waste materials could be able to realize a compost useful for agricultural purposes as an ideal alternative to chemical fertilizers and pesticides.

## Figures and Tables

**Figure 1 plants-10-00386-f001:**
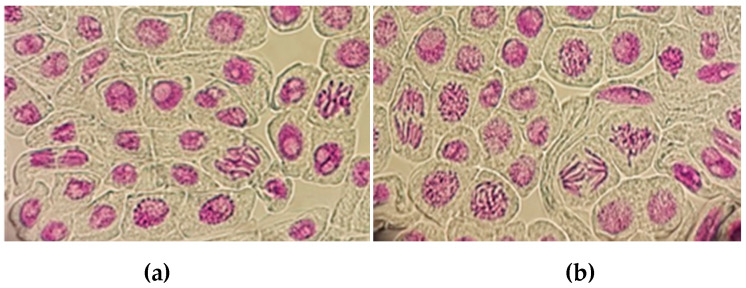
Representative section of root tip cells of broad beans, grown in 0% (**a**) and 20% (**b**) soil sample, showing normal stages (40× resolution).

**Figure 2 plants-10-00386-f002:**
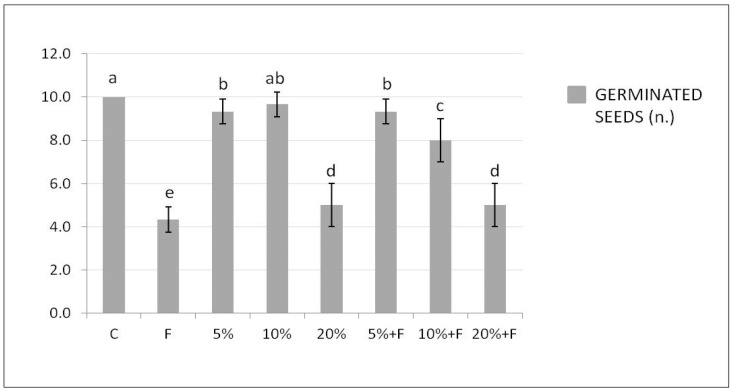
Effect of DdC on tomato seed germination. Bars represent, in order from left to right: control soil untreated and not inoculated with the pathogen (C); soil inoculated with *F. solani* (F); soil treated with DdC at 5%; soil treated with DdC at 10%; soil treated with DdC at 20%; soil treated with DdC at different concentrations and inoculated with the pathogen (5% + F, 10% + F, and 20% + F). All data are expressed as the mean of three replicates ± SDs. Bars with different letters indicate mean values significantly different at *p* ≤ 0.01, according to two-way ANOVA combined with Tukey post hoc test.

**Figure 3 plants-10-00386-f003:**
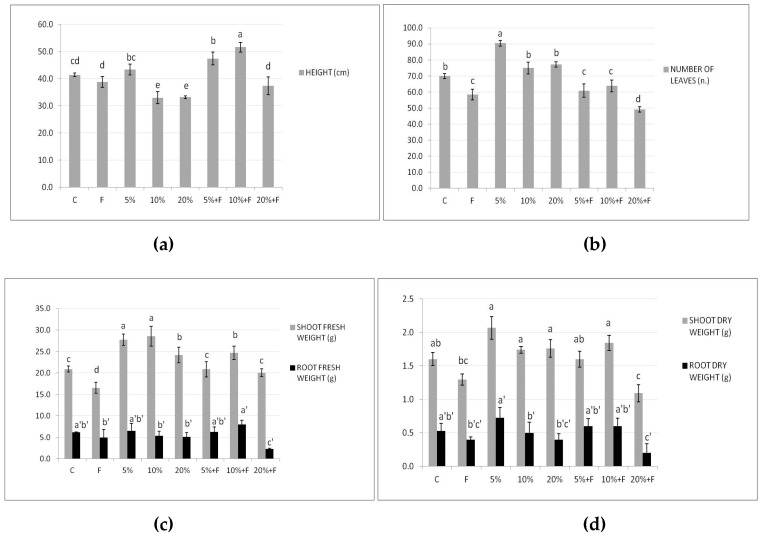
Effect of DdC on tomato growth parameters: height (**a**), number of leaves (**b**), shoot and root fresh weight (**c**), and shoot and root dry weight (**d**). Bars represent, in order from left to right: control soil untreated and not inoculated with the pathogen (C); soil inoculated with *F. solani* (F); soil treated with compost at 5%; soil treated with compost at 10%; soil treated with compost at 20%; soil treated with compost at different concentrations and inoculated with the pathogen (5% + F, 10% + F, and 20% + F). All data are expressed as the mean of three replicates ± SDs. Bars with different letters indicate mean values significantly different at *p* ≤ 0.01, according to two-way ANOVA combined with Tukey post hoc test.

**Table 1 plants-10-00386-t001:** Chemical composition of digestate produced in a zoo-technical farm, and digestate-derived compost obtained in laboratories.

Parameter	Units	Digestate	DdC	L.D. 75/2010 Values ^1^
Moisture (105 °C)	%	91.80 ± 1.2	15.60 ± 0.8	MAC: 50
Ashes	%	2.50 ± 0.1	11.41 ± 1.0	
pH		8.10 ± 0.3	7.88 ± 0.2	6.0 < 8.5
Salinity (Electrical conductivity)	mS/cm	-	3.68 ± 0.1	
Germination index	%	55.00 ± 1.1	59.00 ± 1.5	MRC: 60
Total N	% d.m.	1.04 ± 0.1	9.00 ± 0.3	
Organic N	% d.m.	-	7.63 ± 0.8	
Organic N	% of total N		85.00 ± 2.1	MRC on total N: 80
NH_4_-N	%	-	0.4 ± 0.0	
Total Phosphorus (P)	mg/kg d.m.	350.0 ± 3.2	74.2 ± 1.8	
Total Phosphorus (P_2_O_5_)	mg/kg d.m.	-	169.9 ± 2.3	
Total Phosphorus (P_2_O_5_)	% d.m.	-	0.017 ± 0.0	
Potassium	mg/kg d.m.	-	26,663 ± 6.8	
Potassium (k_2_O)	mg/kg d.m.	-	32,124 ± 4.3	
Potassium (k_2_O)	% d.m.	-	3.2 ± 0.4	
Total organic carbon	% d.m.	6.8 ± 0.2	40.6 ± 0.9	MRC: 20
Carbon (humic + fulvic)	% d.m.	2.5 ± 0.1	11.7 ± 0.2	MRC: 7
C/N ratio		6.5 ± 0.1	4.5 ± 0.2	MAC: 25
*Salmonella* spp. (*n* = 5)	Present or Absent	Absent	Absent	Absent
*Escherichia coli* (n. 1)	MPN g^−1^	1080	438	<1000 CFU g^−1^
*Escherichia coli* (n. 2)	MPN g^−1^	997	352	
*Escherichia coli* (n. 3)	MPN g^−1^	940	304	
*Escherichia coli* (n. 4)	MPN g^−1^	860	287	
*Escherichia coli* (n. 5)	MPN g^−1^	821	258	
Cu	mg/kg d.m.	3.80 ± 0.1	24.70 ± 2.0	MAC: 230.00
Zn	mg/kg d.m.	9.26 ± 0.7	64.11 ± 1.9	MAC: 500.00
Pb	mg/kg d.m.	0.56 ± 0.1	45.70 ± 1.4	MAC: 140.00
Cd	mg/kg d.m.	0.03 ± 0.0	<QL	MAC: 1.5
Ni	mg/kg d.m.	0.24 ± 0.0	3.52 ± 0.2	MAC: 100.00
Hg	mg/kg d.m.	<QL	<QL	MAC: 1.5
Cr (VI)	mg/kg d.m.	<QL	<QL	MAC: 0.5
Al	mg/kg d.m.	<QL	410 ± 3.4	
B	mg/kg d.m.	-	14.9 ± 1.0	
Ba	mg/kg d.m.	-	22.9 ± 0.9	
Fe	mg/kg d.m.	<QL	367 ± 2.1	
Mn	mg/kg d.m.	-	60.8 ± 1.1	
Mo	mg/kg d.m.	-	2.6 ± 0.2	
Sn	mg/kg d.m.	-	2 ± 0.1	
V	mg/kg d.m.	-	0.6 ± 0.0	
Plastic, glass and metals (≥2 mm)	% d.m.	<QL	<QL	MAC: 0.5
Lithoids inerts (≥5 mm)	% d.m.	<QL	<QL	MAC: 5

^1^ Values as reported according to the Italian Legislative Decree 75/2010 about “Reorganization and Revision of the Discipline on Fertilizers”. As, Be, Co, Sb, Se, Ti showed values < detection limits values.MAC = maximum admitted content. MRC = minimum required content. QL = quantifiability limit. MPN = most probable number. CFU = colony forming unit.

**Table 2 plants-10-00386-t002:** Phytotoxic effect of digestate on seed germination, radical elongation and germination index of cress (*L. sativum*) and radish (*R. raphanistrum*).

Species	Treatments	GS ^1^ (n)	RE ^2^ (cm)	GI ^3^ (%)
**Cress**	Control (H_2_O)	11.8 ± 0.3a	9.5 ± 0.4a	98.9 ± 1.3a
	Digestate 2.5%	8.9 ± 0.5ab	8.7 ± 0.5ab	70.0 ± 2.3b
	Digestate 5%	6.4 ± 0.5b	7.1 ± 0.3b	31.7 ± 1.9c
	Digestate 10%	3.2 ± 0.3c	3.8 ± 0.3c	15.8 ± 2.8d
**Radish**	Control (H_2_O)	9.7 ± 0.4a	13.9 ± 0.8a	99.3 ± 2.7a
	Digestate 2.5%	6.8 ± 0.2b	11.0 ± 0.1ab	55.3 ± 3.1b
	Digestate 5%	6.9 ± 0.5b	10.0 ± 0.6ab	44.8 ± 3.2bc
	Digestate 10%	4.6 ± 0.7c	8.1 ± 0.7b	25.8 ± 2.5c

^1^ GS: germinated seed. ^2^ RE; radical elongation. ^3^ GI; germination index. Different letters in the same column per species indicate mean values significantly different at *p* < 0.05 according to one-way ANOVA combined with Tukey post hoc test. Data are expressed as the mean of three replicates ± SDs.

**Table 3 plants-10-00386-t003:** Phytotoxic effect of digestate-derived compost on seed germination, radical elongation and germination index of cress (*L. sativum*) and radish (*R. raphanistrum*).

Species	Treatments	GS ^1^ (n)	RE ^2^ (cm)	GI ^3^ (%)
**Cress**	Control (H_2_O)	9.5 ± 0.6a	11.0 ± 1.2a	99.6 ± 2.5a
	Compost 2.5%	9.0 ± 1.2a	10.8 ± 0.3a	97.6 ± 1.8a
	Compost 5%	7.5 ± 0.6b	9.8 ± 0.5ab	62.1 ± 2.3b
	Compost 10%	5.5 ± 0.4c	7.5 ± 0.6 b	31.2 ± 1.2c
**Radish**	Control (H_2_O)	9.5 ± 0.3a	13.1 ± 0.7a	99.0 ± 1.3a
	Compost 2.5%	9.0 ± 0.8a	12.9 ± 0.3a	94.3 ± 1.5a
	Compost 5%	6.5 ± 0.6b	11.3 ± 0.1ab	63.3 ± 3.2b
	Compost 10%	4.5 ± 0.6c	9.5 ± 0.6b	32.7 ± 2.7c

^1^ GS: germinated seed. ^2^ RE; radical elongation. ^3^ GI; germination index. Different letters in the same column per species indicate mean values significantly different at *p* < 0.05 according to one-way ANOVA combined with Tukey post hoc test. Data are expressed as the mean of three replicates ± SDs.

**Table 4 plants-10-00386-t004:** Antifungal activity of digestate sample.

	Diameter of Inhibition Zones (mm)		
	100%	50%	Azoxystrobin
*A. niger*	0.0 ± 0.0d	0.0 ± 0.0d	15.5 ± 1.2c
*A. ochraceus*	0.0 ± 0.0d	13.2 ± 0.8bc	25.3 ± 1.7bc
*A. flavus*	0.0 ± 0.0d	8.3 ± 1.3c	12.4 ± 2.4c
*M. laxa*	33.3 ± 2.5a	30.5 ± 3.4a	42.6 ± 3.2a
*M.* *fructicola*	34.5 ± 3.5a	31.6 ± 2.9a	41.7 ± 3.5a
*M.* *fructigena*	36.6. ± 3.4a	32.5 ± 2.5a	43.2 ± 3.1a
*F. solani*	33.4 ± 4.2a	31.8 ± 1.8a	41.5 ± 2.7a
*R. solani*	18.7 ± 1.5b	22.4 ± 1.3b	30.2 ± 2.8bc
*S.* *sclerotiorum*	9.8 ± 0.8c	19.6 ± 2.1b	26.2 ± 2.2bc
*B.* *cinerea*	0.0 ± 0.0d	0.0 ± 0.0d	18.4 ± 1.2c
*V. dahliae*	35.6 ± 2.9a	32.2 ± 3.1a	45.6 ± 3.1a

Data were obtained from three replicates ± SDs. Values followed by different letters in the same column indicate mean values significantly different at *p* < 0.05, according to one-way ANOVA combined with Tukey post hoc test.

**Table 5 plants-10-00386-t005:** Antifungal activity of digestate-derived compost sample.

	Diameter of Inhibition Zones (mm)		
	100%	50%	Azoxystrobin
*A. niger*	0.0 ± 0.0c	0.0 ± 0.0c	28.4 ± 2.1c
*A. ochraceus*	0.0 ± 0.0c	0.0 ± 0.0c	25.2 ± 1.2c
*A. flavus*	0.0 ± 0.0c	0.0 ± 0.0c	30.6 ± 3.0bc
*M. laxa*	38.5 ± 3.2a	32.5 ± 2.9a	45.4 ± 3.2a
*M. fructicola*	25.6 ± 2.8b	20.5 ± 1.4b	31.2 ± 1.9bc
*M. fructigena*	0.0 ± 0.0c	0.0 ± 0.0c	45.5 ± 1.5a
*F. solani*	34.3 ± 3.2a	30.2 ± 3.1a	39.3 ± 2.2a
*R. solani*	0.0 ± 0.0c	0.0 ± 0.0c	25.2 ± 2.1c
*S. sclerotiorum*	0.0 ± 0.0c	0.0 ± 0.0c	32.2 ± 3.0bc
*B. cinerea*	0.0 ± 0.0c	0.0 ± 0.0c	38.7 ± 3.5a
*V. dahliae*	38.2 ± 3.1a	35.2 ± 2.7a	42.3 ± 3.4a

Data were obtained from three replicates ± SDs. Values followed by different letters in the same column indicate mean values significantly different at *p* < 0.05, according to one-way ANOVA combined with Tukey post hoc test.

**Table 6 plants-10-00386-t006:** Damping-off (DO) incidence caused by *F. solani* on tomato plants grown in soil inoculated with the pathogen (F) and soil treated with compost at different concentrations (5% + F, 10% + F, and 20% + F).

Treatments	DO (%)
F	56.7 ± 0.6 a
5% + F	6.7 ± 0.6 c
10% + F	26.7 ± 1.0 b
20% + F	53.3 ± 1.0 a

Data are expressed as the mean of three replicates ± SDs. Different letters in the same column indicate mean values significantly different at *p* < 0.01, according to one-way ANOVA combined with Tukey post hoc test.

## Data Availability

The data presented in this study are available on request from the corresponding author.

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
