# Peer review of "Physico-Chemical Characterization and Biological Activities of a Digestate and a More Stabilized Digestate-Derived Compost from Agro-Waste"

_plants, 2021, doi:10.3390/plants10020386_

Round 1

Reviewer 1 Report

Dear authors

I found your paper entitled “Physico-chemical characterization and biological activities of a digestate and a more stabilized digestate-derived compost from agro-wasteof” quality and I suggest publication.

There are some weak points such as Materials and Methods. The hand-made semi-batch reactors should be described in more detail.

And also the Introduction should be improved. There is no research hypothesis, the authors should supplement it.

Conclusions are too long, information from the introduction is repeated. Conclusion should be more conclusive and also generalised confirming if they have wide validity.

Author Response

Dear Reviewer 1,

we have revised the manuscript, according to your comments.

We used the "Track Changes" function in order to highlight all corrections and any revisions, detailed below.

I found your paper entitled “Physico-chemical characterization and biological activities of a digestate and a more stabilized digestate-derived compost from agro-wasteof” quality and I suggest publication.

Authors: Thank you very much for your report and suggestion.

There are some weak points such as Materials and Methods. The hand-made semi-batch reactors should be described in more detail.

Authors: A more detailed description of the hand-made semi-batch reactors has been added in the subsection 3.1 of Material and Methods (lines 354-358), as suggested.

And also the Introduction should be improved. There is no research hypothesis, the authors should supplement it.

Authors: A sentence was added in the introduction section (lines 95-96) to make explicit the research hypothesis.

Conclusions are too long, information from the introduction is repeated. Conclusion should be more conclusive and also generalised confirming if they have wide validity.

Authors: We agree with this comment. Therefore, we have eliminated redundant or unnecessary information in order to make conclusion more generalized and widely valid. 

Reviewer 2 Report

The manuscript by Vitti and colleagues is an important work on the applications of bovine slurry on the growth and development of particular species of agronomic value.

The manuscript is clear and well written, however I believe the authors should frame the manuscript in a more botanical and less agronomic context.

Here are some points that should be explored better

1) Why did the authors choose these species: Lepidium sativum, Raphanus raphanistrum, Solanum Lycopersicum and Vicia vaba? The authors should explain why they chose these species, is there a taxonomic link between these species? Did they choose them only for their agronomic importance?

2) Why did the authors perform the genotoxicity test only on fava bean plant?

3) Why did the authors evaluate the effects of F. solani in the tomato plant?

Author Response

Dear Reviewer 2,

we have revised the manuscript, according to your comments.

We used the "Track Changes" function in order to highlight all corrections and any revisions, detailed below.

The manuscript by Vitti and colleagues is an important work on the applications of bovine slurry on the growth and development of particular species of agronomic value.

The manuscript is clear and well written, however I believe the authors should frame the manuscript in a more botanical and less agronomic context.

Authors: The authors are very grateful for the comment. Effectively, the context of this study is designed based on searching of eco-friendly products as an ideal alternative to the chemical fertilizers and pesticides. Therefore, for the authors, this is a topic of predominantly agronomic interest, with an important point of view at the low environmental impact and circular economy. However, for a correct botanical clarification, the family has been added for each species mentioned in the manuscript.

Here are some points that should be explored better

1) Why did the authors choose these species: Lepidium sativum, Raphanus raphanistrum, Solanum Lycopersicum and Vicia vaba? The authors should explain why they chose these species, is there a taxonomic link between these species? Did they choose them only for their agronomic importance?

Authors: The tested seeds have been often used in this kind of phytotoxic assays because of their easy and well-known germinability. In addition, the tested plants used in the phytotoxic assay Lepidium sativum, Raphanus raphanistrum and Solanum Lycopersicum are fast-growing plant and are popular in different zones like Asia, Europe and USA and highly used in the rural area. In particular, L. sativum has been used often in phytotoxic assays because of its sensitivity to the toxic factors of pharmaceutical drugs, whereas R. raphanistrum  is  sensitive to most allelochemicals and, finally, the chromosomal aberrations and micronuclei induced in Vicia faba by genotoxins are widely used to study environmental pollution.

Several scientific research have considered the use of plants as indication of the phytotoxicity and genotoxicity of different treatments, as following:

  1. Mecina G.F., Santos V.H.M., Dokkedal A.L., Saldanha L.L., Silva L.P., Silva R.M.G.2014. Phytotoxicity of extracts and fractions of Ouratea spectabilis (Mart. ex Engl.) Engl. (Ochnaceae). South African Journal of Botany.95, 174-180
  2. Della Pepa T., Elshafie H.S., Capasso R., De Feo V., Camele I., Nazzaro F., Scognamiglio M.R. and Caputo L. 2019. Antimicrobial and phytotoxic activity of Origanum heracleoticum and O. majorana essential oils growing in cilento (Southern Italy). Molecules, 24, 2576; 1-16.
  3. Gruľová, D., Caputo, L., Elshafie, H. S., Baranová, B., De Martino, L., Sedlák, V., Camele I. and De Feo, V. 2020. Thymol chemotype Origanum vulgare L. essential oil as a potential selective bio-based herbicide on monocot plant species. Molecules, 25(3), 595.
  4. Rolim de Almeida L.F., Frei F., Mancini E., De Martino L. and De Feo V., 2020. Phytotoxic Activities of Mediterranean Essential Oils. Molecules 2010, 15, 4309-4323
  5. Marcato-Romain C.E., Guiresse M.,, Cecchi M., Cotelle S., Pinelli E.(2009). New direct contact approach to evaluate soil genotoxicity using the Vicia faba micronucleus test. Chemosphere 77:345–350

2) Why did the authors perform the genotoxicity test only on fava bean plant?

Authors: Vicia faba is a model test plant commonly used in the detection and diagnosis of genotoxicity of pollutants since 1982, when the micronucleus test in this species was established by Ma and by Degrassi and Rizzoni. The tests routinely employed include the assay in root tips. No genotoxicity effect was induced by our compost in terms of frequencies of micronuclei on this model plant. Otherwise, we would surely have considered some other species as well in addition to Vicia faba.  

  1. Ma TH (1982) Vicia cytogenetic tests for environmental mutagens. A report of the U.S. Environmental Protection Agency Gene-Tox Program. Mutat Res 99:257–271
  2. Degrassi F, Rizzoni M (1982) Micronucleus test in Vicia faba root tips to detect mutagen damage in fresh-water pollution. Mutat Res 97:19–33

3) Why did the authors evaluate the effects of F. solani in the tomato plant?

Authors: This pathogenic soil-borne fungus represents one of the main causal agent of damping-off in tomato, which is considered one of the most important and widely consumed vegetables worldwide. A suppressive action against F. solani in tomato plants by a compost was never considered before this study, and therefore intended by authors as an interesting challenge in the context of sustainable disease control.

Reviewer 3 Report

The research presented in this study is very interesting and well curated. The topic of compost and the possibilities of replacing chemical fertilizer is very current, so this research adds to the study useful for this purpose. The manuscript is well written, very understandable and the study followed the scientific steps well. The paragraph on materials and methods should be moved before the results. I think the results are generally more understandable if you first learn how they were obtained. Furthermore, the values of table 1 relating to the characteristics of the digestate do not report standard deviation. Were there no repetitions in the sampling? If so, it would be better to highlight the stability of the values obtained.

Author Response

Dear Reviewer 3,

we have revised the manuscript, according to your comments.

We used the "Track Changes" function in order to highlight all corrections and any revisions, detailed below.

The research presented in this study is very interesting and well curated. The topic of compost and the possibilities of replacing chemical fertilizer is very current, so this research adds to the study useful for this purpose. The manuscript is well written, very understandable and the study followed the scientific steps well.

Authors: Thank you very much for your report and accurate reading of the manuscript.

The paragraph on materials and methods should be moved before the results. I think the results are generally more understandable if you first learn how they were obtained.

Authors: Results section precedes the Materials one to comply with the formatting required by the journal.

Furthermore, the values of table 1 relating to the characteristics of the digestate do not report standard deviation. Were there no repetitions in the sampling? If so, it would be better to highlight the stability of the values obtained.

Authors: Thank you very much for the accurate review of this table. We have added standard deviations, as  rightly suggested.